# The Effect of Acute Caffeine Ingestion on Tactical Performance of Professional Soccer Players

**DOI:** 10.3390/nu14071466

**Published:** 2022-03-31

**Authors:** Rodrigo Freire de Almeida, Israel Teoldo da Costa, Guilherme Machado, Natalia Madalena Rinaldi, Rodrigo Aquino, Jason Tallis, Neil David Clarke, Lucas Guimaraes-Ferreira

**Affiliations:** 1Postgraduate Program in Physical Education, Center of Physical Education and Sports, Federal University of Espirito Santo, Vitória 29075-910, Brazil; rodrigofalmeida80@gmail.com (R.F.d.A.); natalia.rinaldi@ufes.br (N.M.R.); rodrigo.aquino@ufes.br (R.A.); ad6463@coventry.ac.uk (L.G.-F.); 2Centre of Research and Studies in Soccer (NUPEF), Physical Education Department, Federal University of Viçosa, Viçosa 36570-900, Brazil; israel.teoldo@ufv.br (I.T.d.C.); machado.guilhermef@gmail.com (G.M.); 3EDAF Research Group, Faculty of Education, Universidad de Castilla-La Mancha (UCLM), 02071 Albacete, Spain; 4Centre for Sport, Exercise and Life Sciences, Coventry University, Coventry CV1 5FB, UK; jason.tallis@coventry.ac.uk; 5School of Life Sciences, Coventry University, Coventry CV1 5FB, UK

**Keywords:** supplementation, soccer, decision-making, tactical performance

## Abstract

In soccer, physical, tactical, and decision-making processes are highly important facets of successful performance. Caffeine has well established effects for promoting both physical and cognitive performance, but the translation of such benefits specifically for soccer match play is not well established. This study examined the effects of acute caffeine ingestion on tactical performance during small-sided games (SSG) in professional soccer players. Nineteen soccer players (22 ± 4 years) underwent a randomized, counterbalanced, crossover, double-blind placebo-controlled trial. The protocol consisted of 5 bouts of 5-min SSG with 3 players plus a goalkeeper in each team (3 + GK × 3 + GK) with each SSG separated by 1 min rest intervals. Tactical performance was assessed using the system of tactical assessment in soccer (FUT-SAT). Prior to each experimental trial, participants ingested caffeine (5 mg·kg^−1^) or a placebo 60 min before the protocol. Overall, caffeine ingestion resulted in an increased ball possession time when compared to the placebo. When the offensive and defensive core principles were analyzed, the results were equivocal. Caffeine resulted in positive effects on some tactical decisions during the protocol, but it was deleterious or promoted no observed effect on other of the core tactical principles. Caffeine ingestion resulted in less offensive (during SSG3) and defensive (SSG 2, SSG3, and SSG4) errors. Caffeine ingestion also resulted in higher total offensive success during SSG 1 and SSG2, but it was detrimental during SSG3. Additionally, total defensive success was lower for the caffeine conditions during SSG 2 and SSG5 when compared to the placebo. In conclusion, caffeine influenced aspects of tactical decisions in soccer, resulting in fewer offensive and defensive errors, although it may be deleterious considering other tactical parameters. Future studies may clarify the effects of caffeine ingestion on specific decision-making parameters in soccer.

## 1. Introduction

The performance-enhancing effects of acute caffeine ingestion have been explored at length, with well established benefits for endurance, intermittent, and resistance exercise [1,2] as well as low order cognitive functions [3]. More specifically, caffeine has been shown to improve physical and technical elements needed for successful soccer match play, with evidence indicating improved repeated sprint and jump performance [4], reactive agility [5], and passing accuracy [6]. Such effects have likely led to the high prevalence of caffeine use in professional soccer, where 97% of sampled English professional soccer teams are providing caffeine to players to improve performance [7].

Team sports, such as soccer, are open skill activities where athletes need to process in-game information and respond quickly with speed and accuracy, and the ability to deal successfully with such processes is often referred to as “soccer IQ” [8]. Thus, soccer is multifaceted in its neurophysiological demand, requiring perceptual-cognitive and perceptual-motor skills, both of which contribute to the achievement of high-performance levels [9].

The majority of caffeine studies in soccer assessed performance using physical tests such as sprinting, vertical jumps, repeated sprints, and change of direction performance [10,11], which are important physical facets of soccer match play. Replicating the demands of soccer match play in repeated measures designs is challenging, and there is a dearth of studies examining the effect of caffeine during more ecologically valid in-game scenarios. Despite this, a small number of studies have examined the effects of caffeine ingestion on soccer player movement during simulated games [10,12]. For example, Del Coso et al. [10] demonstrated that the ingestion of a caffeine-containing energy drink resulted in higher total distance covered at medium-intensity running (8.1–13.0 km·h^−1^), high-intensity running (13.1–18.0 km·h^−1^), and sprinting (more than 18.0 km·h^−1^) when compared to a caffeine-free drink (*p* < 0.05). For a better understating of the potential of caffeine as an ergogenic aid for the soccer athlete, work is needed to determine the effect of caffeine in more complex and ecological environments with specific game situations involving decision-making situations [13].

To address this gap in the literature, the present study investigated the effect of acute caffeine ingestion (5 mg·kg^−1^) on the tactical performance of professional soccer players during small-sided games. It was hypothesized that acute caffeine ingestion would improve tactical performance in small-sided games.

## 2. Materials and Methods

### 2.1. Subjects

Nineteen male professional soccer players participated in the study (22 ± 4 years and 75.6 ± 5.7 kg of body mass). An a priori power calculation (G Power; v 3.1.4) for a 2 factor repeated measures ANOVA, based on an estimated observed power (1-β) of 0.85, an estimated effects size of 1, with an alpha set at 0.05 indicated a sample size of 10 participants per condition was needed for the study. There is no consensus on the accepted level of 1-β, but values between 0.8 and 0.9 are commonly used for sample size estimation [14]. Forty-seven players were available in the squad and initially recruited, however, twenty-eight dropped out. Twenty-four players were excluded by club request (i.e., precautions due to a history of injuries, or other commitments during test days), three were due to injuries, and one chose not to participate in the study. The final sample consisted of 19 participants, including starters and reserves of the club’s first team.

Participants were all professional soccer players from the Brazilian fourth division. The institution’s Human Research Ethics Committee approved the procedures used in this study (protocol number: 66849317.9.0000.5542) and all the athletes provided written informed consent. The study was conducted in accordance with the 2008 Declaration of Helsinki and the resolution of the Brazilian National Health Council.

### 2.2. Design and Procedures

A randomized, counterbalanced, crossover, double-blind placebo-controlled experimental design was used (Figure 1). All participants performed one familiarization session one week before the experimental sessions, which followed the experimental trial procedures. Initially, participants rested in a seated position for 10 min to acquire a resting heart rate (HR) assessed via telemetry (Geonaute CR2032 OnRhythm 50^®^, Decathlon Ltd., Villeneuve d’Ascq, France). Then, the participants ingested 500 mL of cold flavored solution (a non-caloric juice powder) with caffeine (5 mg/kg^−1^; Sigma-Aldrich^®^, Burlington, MA, USA) or a flavored solution alone, which was used as a placebo. Solutions were considered identical in flavor and color by three researchers involved in the study. Daily caffeine consumption was assessed using a 7-day caffeine recall based on the caffeine content in common food and beverages, according to Maughan [15]. The mean daily caffeine intake was 119.6 ± 136.7 mg (ranging from 0 to 439.1 mg).

Tests were conducted the day after the players’ rest day, and participants were asked to abstain from intense exercise at least 24 h prior to the trials. Participants were also asked to follow the same diet and exercise practices before each trial and abstain from caffeine consumption (in drinks and supplements) 24 h prior to testing sessions. Both experimental sessions were performed in the morning starting at 8:00 am to avoid variations in the circadian cycle [16] with an interval of 1 week between sessions. The blinding protocol was assessed by a model adapted from Klauss et al. [17]. This model consists of two questions: (1) Do you think you have received/are receiving treatment? Regardless of the answer, yes or no, the second question was asked; (2) How confident is your impression? This last question contained a Likert scale from 1 to 5 (1: none; 2: little; 3: average; 4: much and 5: extreme). Only 16% (*n* = 2) of the participants correctly identified the use of caffeine in this condition. The Likert scale on how confident participants were about the treatment received, indicated a median value of 3 for both conditions (placebo and caffeine).

Sixty minutes following the consumption of caffeine or placebo solutions, participants performed a generic warm-up consisting of 10 bodyweight squats, 10 forward lunges on each side, and 3 min of dynamic stretching of relevant lower limb musculature, followed by a protocol consisting of five bouts of 5-min small-sided games (SSG) separated by 1 min of rest. Rest periods were used for HR measurements and the rating of perceived exertion (RPE) using a 0–10 scale [18], and ad libitum hydration. The complete protocol consisted of 30 min of activity.

Environmental conditions were measured using a digital thermo-higro-anemometer (Akrom KR825^®^, São Leopoldo, RS, Brazil). Temperature session 1: 29.6 °C, session 2: 29.45 °C. Relative humidity: session 1: 48%, session 2: 42%; wind speed session 1: 1.8 m/s, session 2: 0.9 m/s.

### 2.3. Tactical Performance

The tactical performance was assessed using the System of Tactical Assessment in Soccer (FUT-SAT), as described by [19,20]. The FUT-SAT protocol enables the assessment of tactical actions performed by players with and without ball possession, either in offensive or defensive situations, near or distant from the ball, according to the ten core tactical principles of soccer (see Table 1) [20,21], and presents both an intra and inter-observers’ reliability higher than 0.79 [22]. The protocol consisted of 5 SSG and each SSG consisted of 2 teams with 4 players (3 outfield players + goalkeeper × 3 outfield players + goalkeeper). The SSG was played on a natural grass pitch with a field size of 36 m long by 27 m wide. The game was played according to the official FIFA rules of soccer, and an experienced soccer coach performed the referee’s role. Before the session, players were asked to perform at their maximum during the games, and no feedback was provided. Team formations and experimental conditions (caffeine and placebo) were randomized. Each team consisted of a goalkeeper (not used in the analysis), a defender, a midfielder and a forward. The team formation was the same for both experimental trials, and all players of each team were under the same condition (caffeine or placebo).

All SSG were recorded using a digital camera (Sony NEX-F3K–16.1 Megapixel, SONY^®^, Manaus, Brazil) positioned diagonally, at least 5 m away and 8 m high. Video processing and analysis were performed using the Soccer Analyzer software. This software was developed for use with FUT-SAT and enables the insertion of spatial references and the accurate verification of the position and movement of the players, as well as the analysis and categorization of the actions that were to be assessed. Forty-five games were played, summing 225 min (13,500 s). A total of 21.906 tactical actions (offensive: 12.739 and defensive: 9.167) were performed by the players.

The tactical performance index provided by the output of FUT-SAT is based on the number, quality, place, and results of tactical actions (for more details, please see Costa et al. [22]). Tactical success considers the number of correct actions performed divided by the total number of actions for the respective phase of play. The dependent variables are: (i) variables inherent to the exercise protocol: HR; RPE; (ii) ball possession; (iii) core tactical principles of soccer (as shown in Table 1). Video analysis was carried out by an experienced researcher, as described by Costa et al. [22,23].

### 2.4. Statistical Analysis

Data were presented as mean ± standard deviation and by the 95% confidence interval. Data normality was assessed via the Shapiro–Wilk test. As data were normally distributed, two-way ANOVA with repeated measures was used (condition: caffeine × placebo vs. time: SSG1 to SSG5). Post hoc analysis using Bonferroni adjustments was performed where any significant interactions and main effects were found. The level of significance used was 5% (*p* < 0.05). GraphPad Prism software, version 8.0 (GraphPad Software, Inc., San Diego, CA, USA) was used for the statistical analysis and figure generation.

The effect sizes (ES) were estimated by the magnitude-based inference (MBI) calculated by the spreadsheet provided by Hopkins et al. [24] to capture the effects of the chances of minimum difference consolidating the significance of differences between conditions. Thus, the practical implications were classified as having a beneficial, negligible, or deleterious effect. The percentage scores present in these data were classed as follows: <1% almost certainly not; 1–5% very unlikely; 5–25% unlikely; 25–75% possibly; 75–95% likely; 95–99% very likely; >99% almost certain.

## 3. Results

### 3.1. Physiological and Subjective Performance

No differences in HR were observed between the caffeine and placebo conditions (F1,18: 0.785; *p* = 0.387). A significant main effect was observed for time (F5,90: 472.4; *p* < 0.0001), with no interaction for the main effect (F5,90: 1.465; *p* = 0.21). Post hoc analysis and MBI indicated a higher HR than the caffeine condition (*p* = 0.99; ES: 0.37; “likely deleterious”) during rest. No differences were observed in RPE between conditions (F1,18: 1.107; *p* = 0.31) but a significant effect of time (SSGs) was observed (F4,72: 80.42; *p* < 0.0001) and no interaction (F4,72: 0.997; *p* = 0.41) (Figure 2).

### 3.2. Tactical Performance

As shown in Figure 3A, total ball possession was higher in the caffeine condition compared to the placebo condition (*p* = 0.023; ES: 0.56; “likely beneficial”). When each SSG was analyzed (Figure 3B), ball possession was higher for caffeine compared to the placebo in SSG2 (ES: 0.83; “likely beneficial”); SSG3 (ES: 0.96; “likely beneficial”); and SSG5 (ES: 0.81; “likely beneficial”).

Table 2 presents the outcomes related to the offensive phase of the core tactical principles of soccer and the effects of caffeine based on MBI. For the penetration offensive principle, caffeine presented a “likely beneficial” effect in tactical efficiency and is “possibly beneficial” for the number of errors, compared to the placebo. For the offensive unity, the results are mixed. caffeine resulted in a “likely beneficial” effect in frequency, but “likely deleterious“ for efficiency and error in this offensive core principle. caffeine was also “possibly beneficial” for the frequency and number of errors in the offensive core principle width and length without the ball.

The defensive core tactical principles in response to caffeine and placebo ingestion are presented in Table 3. caffeine resulted in “likely beneficial” effects on the number of errors in defensive coverage and concentration, and tactical efficiency and error on delay and defensive unity core principles. However, MBI indicates a “likely deleterious“ or “very likely deleterious“ effect of caffeine on the tactical performance index, frequency, and tactical efficiency for the defensive coverage core principle. Additionally, this indicated a “possibly deleterious“ effect for frequency in delay, and “likely deleterious” for frequency in the concentration principle.

Total offensive and defensive tactical behavior were calculated from core principles data and are presented in Figure 4. For the total of offensive success (TOS) no difference was observed between the conditions (F1,18: 1.59; *p* = 0.223), but a main effect was observed for time (F4,72: 3.02; *p* = 0.023). No interaction was observed (F4,72: 0.992; *p* = 0.415). According to MBI, higher performance in the placebo condition was found compared to caffeine in SSG3 (*p* = 0.999; ES: −0.45; “likely deleterious”; Figure 4A). No significant differences were observed for the total of offensive errors (TOE) between the conditions (F1,18: 0.17; *p* = 0.682), for time (F4,72: 0.53; *p* = 0.711) or interaction (F4,72: 1.00; *p* = 0.413). Players in the placebo condition presented more offensive errors in SSG3 according to MBI (ES: −0.42; “likely beneficial”; Figure 4B).

The total of defensive success (TDS) was also analyzed. A main effect was observed for the conditions (F1,18: 5.04; *p* = 0.037), without differences across time (F4,72: 1.91; *p* = 0.117) or interaction (F4,72: 1.56; *p* = 0.195). According to MBI, meaningful differences were detected between conditions with a higher performance in the caffeine condition during SSG2 (*p* = 0.999; ES: −0.38; “likely beneficial”) and SSG5 (*p* = 0.9; ES: −0.65; “likely beneficial”) compared to the placebo condition (Figure 4). Conversely, no differences between the caffeine and placebo conditions were found in the total of defensive errors (TDE) (F1,18: 1.55; *p* = 0.223), although significant main effects for time (F4,72: 4.49; *p* = 0.002) and interaction (F4,72: 3.40; *p* = 0.013) were observed. Differences emerged according to MBI, pointing to more errors made by players in the placebo condition during SSG2 (*p* = 0.235; ES: −0.63; “very likely beneficial”); SSG3 (*p* = 0.999; ES: −0.41; “likely beneficial”) and SSG4 (*p* = 0.999; ES: −0.79; “very likely beneficial”, as presented in Figure 4D.

## 4. Discussion

The current study investigated the effect of acute caffeine ingestion on tactical performance in professional soccer players. Overall, players exhibited lower defensive errors under caffeine conditions when compared to the placebo. A higher ball possession was also observed in the caffeine condition. When the offensive and defensive core principles were analyzed individually, the results were equivocal. Caffeine resulted in positive effects on some tactical decisions during the protocol, but it was deleterious or promoted no observed effect in other of the core tactical principles. For example, total defensive success was lower in the caffeine condition compared to the placebo during SSG2 and SSG5.

In this study, the variables related to physiological and subjective performance showed similar outcomes between the conditions. With regards to tactical performance, the players in the caffeine condition presented better scores in some offensive variables like penetration. For offensive unity, players in the placebo condition performed better. This resulted in a similar total offensive success among the conditions, except for SSG3, where players in the placebo condition presented a better score compared to the caffeine condition.

During the defensive phase, placebo conditions have a better outcome for the coverage principle. Game by game, the placebo’s performance was higher for total defensive success during SSG2 and SSG5 when compared to caffeine’s performance. The defensive predominance of players in the placebo condition may be explained by the higher values of ball possession among players in the caffeine condition, as pointed out by the MBI analysis. However, in delay and defensive unity, more tactical errors were committed by the players in the placebo condition in SSG3, SSG4 and SSG5. Similarly, caffeine resulted in fewer errors during the offensive phases of the game, but only in SSG3. These results may indicate that players in caffeine conditions are more precise in tactical actions, which supports the ergogenic effect of caffeine on professional soccer players. However, the implications of differences in ball possession time between conditions and how this might impact offensive and defensive success rates remain to be further determined. It is worth mentioning, that during small-sided games, the total number of ball disputes and loss of ball possession was greater than in the actual games for all player positions [25].

Several cognitive functions are involved in tactical actions in the soccer game. Previous studies have shown decision errors committed by players during the game are related to control and executive functions [26]. The appropriate motor or emotional actions and the inhibition of the same inappropriate actions, in certain contexts, go through this cognitive aspect. The treatment of relevant sensory information (e.g., visual information) detected in the environment is fundamental to the general volitional decision-making process [27,28,29,30], as well as in soccer [31,32,33]. Evidence points to a decrease in passing technical accuracy after 15 min in small-sided games (5 × 5) [34] and 90 min in the formal game [30]. The results observed in the current study corroborate this finding. However, the literature shows a drop in accuracy in specific decision-making tests [35]. Recent findings demonstrated that, for mental fatigue, soccer players decrease their peripheral perception, make more decision-making errors in-game, and have a compensatory increase in physical attrition [35]. Overall, the literature shows a negative mental fatigue effect in physical, technical, tactical, and cognitive aspects related to decision-making in soccer [33,35]. It is justified that fatigue, highlighted in the aforementioned studies, such as tiredness, a lack of energy, changes in mood, impaired reaction time, inattention, and drop accuracy [36] are related to the drop in specific decision-making [30,35,37].

It has already been shown that caffeine intake can be beneficial for the physical performance of soccer athletes [10,11,12]. However, a soccer game goes far beyond physical performance, and the search for strategies that also lead to improvements in the cognitive and decision-making aspects during the game can be of great importance for these athletes. Coaches and sports nutritionists may consider the specifics of each position to determine the use of caffeine supplementation in their athletes. For example, based on the results herein, it could be more beneficial for forwards compared to defenders. A challenge, when interpreting the results of the present study, is the identification of the reasons explaining the mix of positive, neutral and deleterious effects of acute caffeine ingestion on different tactical principles of soccer. One possibility is that different levels of arousal are needed for different aspects of decision-making during the game. The Yerkes–Dobson law states that performance increases with higher levels of arousal up to a certain point and then decreases, in an “inverted U shape” [38]. The effects of caffeine on arousal levels are known [39,40], but the importance of different arousal levels in different tactical and decision-making actions in soccer is not yet fully understood. Although speculative, one hypothesis is that different levels of arousal are required for different tactical actions during the game and that the effects of caffeine are different according to this relationship. This is an area of interest for future studies.

To the best of our knowledge, this is the first study to investigate the effect of ergogenic aids on the tactical performance of professional soccer players. Nevertheless, this study has limitations. For example, ball possession time was longer in the caffeine condition when compared to the placebo. It was not possible to establish whether ball time exerts any relationship with tactical variables during the game and whether differences observed in offensive and defensive success between conditions were to some extent due to differences in ball time. This was a challenge, since when using experimental designs that include real games, it becomes difficult, or even impossible, to separate these variables. The experimental design used herein presents great ecological validity, but at the same time limits the understanding of the real effects of caffeine ingestion in each tactical aspect. A possibility for future research is to combine laboratory-based data with in-game tactical-decision data to access whether specific effects of caffeine ingestion (e.g., improved reaction time or increased arousal) could be related to specific tactical decision-making elements related to offensive and defensive success, as assessed in this study. Additionally, in the current study, the blinding protocol seems to have been effective. However, the possibility that the expectancy of the caffeine effects may have influenced some players cannot be discarded. Shabir et al. [41] demonstrated that the expectation of the effects of caffeine on physical and cognitive performance can influence the ergogenic effects of caffeine. For example, Saunders et al. [42] observed that the correct identification of caffeine ingestion resulted in a slightly greater effect of this supplement on a 1 h cycling time-trial performance compared to placebo and control conditions. In any case, as only two players correctly identified the use of caffeine, the outcome of the randomized trial does not seem to have been negatively affected. It is also important to note, that inconsistencies observed between studies with caffeine supplementation may be due to inter-individual variability in the biological response [43,44], which was not assessed in this study. The current research opens new possibilities for the understanding of caffeine ergogenic effects.

## 5. Conclusions

The acute intake of 5 mg·kg^−1^ of caffeine, 1 h before SSG, influenced the tactical performance of professional players, mainly by reducing the number of decision-making errors during the protocol. However, in several time points, caffeine ingestion resulted in lower tactical parameters and total defensive success. However, this is only the first study to assess the influence of caffeine on tactical performance and the results should be interpreted with caution. Since positive effects on physical performance in response to caffeine intake are well established, this aspect must also be considered for the use of caffeine by soccer players. Future studies could contribute to a better understanding of the effects of caffeine intake on tactical performance and decision-making in soccer.

## Figures and Tables

**Figure 1 nutrients-14-01466-f001:**
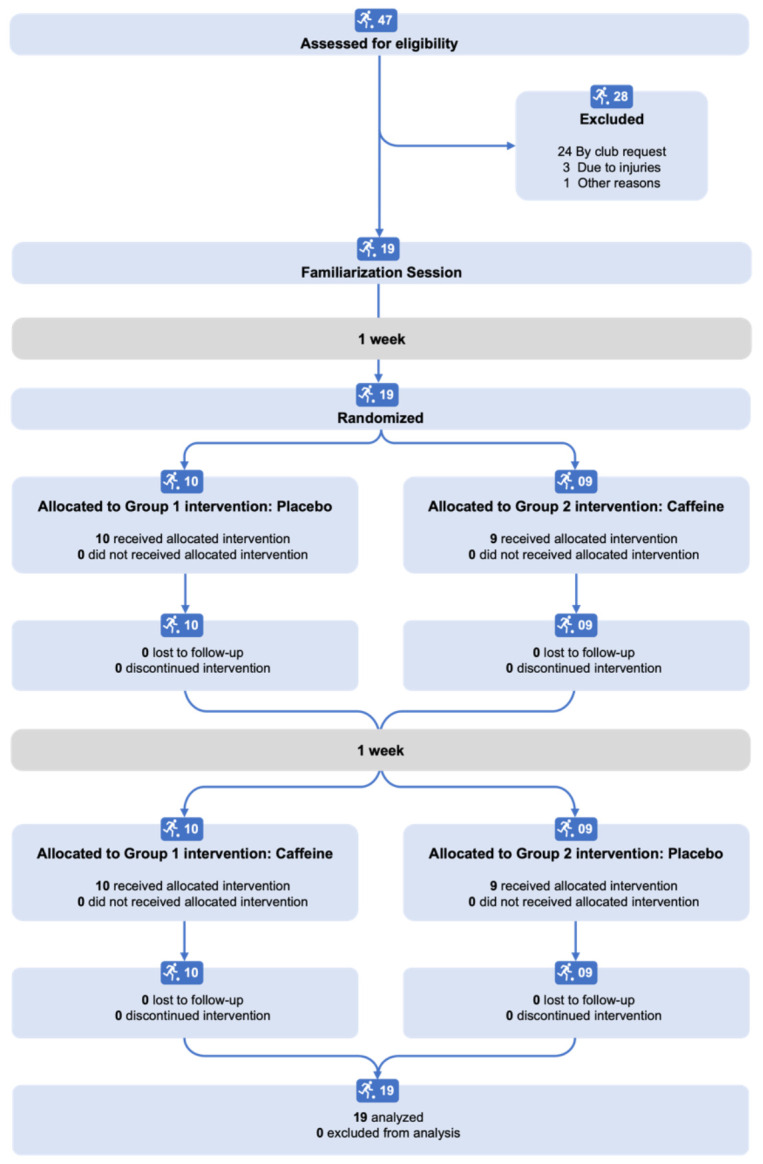
Flow diagram illustrating the study design.

**Figure 2 nutrients-14-01466-f002:**
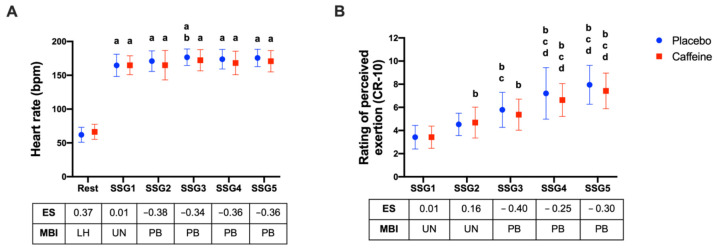
(**A**) Heart rate at rest and after five SSG; (**B**) rating of perceived exertion after five SSG. (a) *p* < 0.05 vs. rest; (b) *p* < 0.05 vs. SSG1; (c) *p* < 0.05 vs. SSG2; (d) *p* < 0.05 vs. SSG3. AU: arbitrary unit; ES: effect size; MBI: magnitude-based inference; PB: possibly beneficial; PO: possibly; UN: unclear; LH: likely deleterious; *p*: *p*-value; SSG1: game 1; SSG2: game 2; SSG3: game 3; SSG4: game 4; and SSG5: game 5.

**Figure 3 nutrients-14-01466-f003:**
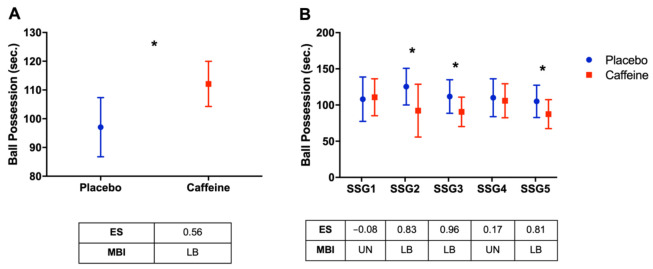
(**A**) Ball possession (total) in seconds; (**B**) ball possession in each SSG in seconds. * *p* < 0.05 placebo vs. caffeine. AU: arbitrary unit; ES: effect size; MBI: magnitude-based inference; UN: unclear; LB: likely beneficial; *p*: *p*-value; SSG1: game 1; SSG2: game 2; SSG3: game 3; SSG4: game 4; and SSG5: game 5.

**Figure 4 nutrients-14-01466-f004:**
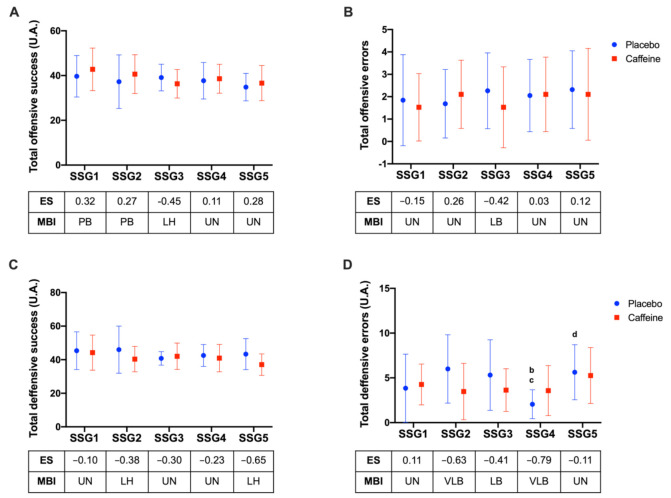
(**A**) Total offensive success; (**B**) total offensive errors; (**C**) total defensive success; (**D**) total defensive errors; (a) (*p* < 0.05) differences to rest; (b) to SSG1; (c) to SSG2; (d) to SSG3. AU: arbitrary unit; ES: effect size; MBI: magnitude-based inference; PB: possibly beneficial; UN: unclear; LB: likely beneficial; LH: likely deleterious; VLB: likely beneficial; *p*: *p*-value; SSG1: game 1; SSG2: game 2; SSG3: game 3; SSG4: game 4; and SSG5: game 5.

**Table 1 nutrients-14-01466-t001:** Description of the core tactical principles of soccer ^1^.

Phases of Play	Principles	Description
Offensive	Penetration	Reduction of the distance between the player in possession of the ball and the opponent’s goal or goal line.
	Offensive Coverage	Providing offensive support to the player in possession.
	Depth Mobility	Generation of organizational instability in the opposing defense.
	Width and Length without the ball	Utilization and increase of the effective play-space in width and depth.
	Offensive Unity	Progression movements or offensive support by the player (s) who compose (s) the last transversal line (s) of the team.
Defensive	Delay	Opposition to the player in possession.
	Defensive Coverage	Providing defensive support to the player performing delay.
	Balance	Numerical stability or superiority in opposition relations.
	Concentration	Increase of defensive protection within the riskier zone to the goal.
	Defensive Unity	Reduction of the opposition’s effective play-space.

^1^ Source: Teoldo et al. [20].

**Table 2 nutrients-14-01466-t002:** Results of the core tactical principles of soccer featuring offensive principles.

Phase	Core Tactical Principles	Principle Location	Detailed Performance	Placebo (ME ± SD)	Caffeine (ME ± SD)	MBI-Value (Classification)
**Offensive Principles**	Penetration	ICP	TPI	45.38 ± 21.62	45.45 ± 18.13	0.01 (unclear)
			Frequency	2.98 ± 2.03	2.86 ± 1.63	−0.07 (unclear)
			% Tactical Efficiency	78.81 ± 32.71	86.59 ± 25.68	0.46 (likely beneficial)
			Errors	0.36 ± 0.61	0.25 ± 0.45	−0.44 (possibly beneficial)
	Offensive Coverage	ICP	TPI	46.87 ± 15.58	46.27 ± 18.04	−0.08 (unclear)
			Frequency	5.32 ± 3.09	4.65 ± 2.95	−0.39 (unclear)
			% Tactical Efficiency	92.60 ± 20.20	90.58 ± 25.78	−0.15 (unclear)
			Errors	0.27 ± 0.70	0.20 ± 0.69	−0.18 (unclear)
	Width and Length without the ball	OCP	TPI	47.60 ± 6.74	48.05 ± 6.35	0.11 (unclear)
			Frequency	21.52 ± 6.74	22.75 ± 7.97	0.26 (possibly beneficial)
			% Tactical Efficiency	96.12 ± 5.49	97.17 ± 5.13	0.32 (unclear)
			Errors	0.83 ± 1.15	0.57 ± 0.96	−0.38 (possibly beneficial)
	Depth Mobility	OCP	TPI	48.69 ± 29.67	44.41 ± 30.54	−0.23 (unclear)
			Frequency	1.52 ± 1.30	1.55 ± 1.30	0.04 (unclear)
			% Tactical Efficiency	79.12 ± 40.23	72.37 ± 43.59	−0.26 (unclear)
			Errors	0.04 ± 0.32	0.07 ± 0.33	0.21 (unclear)
	Offensive Unity	OCP	TPI	46.01 ± 14.40	47.57 ± 15.56	0.19 (unclear)
			Frequency	6.23 ± 3.49	7.29 ± 3.70	0.57 (likely beneficial)
			% Tactical Efficiency	90.83 ± 15.37	86.92 ± 20.68	−0.53 (likely deleterious)
			Errors	0.55 ± 0.84	0.79 ± 1.10	0.53 (likely deleterious)

Note: % Tactical efficiency: percentage of tactical efficiency; MBI: magnitude-based inference; errors: tactical errors; ICP: inside the center of play; ME ± SD: mean and standard deviation; OCP: outside the center of play; SSG: small-sided game; TPI: tactical performance index.

**Table 3 nutrients-14-01466-t003:** Results of the core tactical principles of soccer featuring defensive principles.

Phase	Core Tactical Principles	Principle Location	Detailed Performance	Placebo (ME ± SD)	Caffeine (ME ± SD)	MBI-Value (Classification)
**Defensive Principles**	Delay	ICP	TPI	34.31 ± 11.73	33.66 ± 10.96	−0.13 (unclear)
			Frequency	6.84 ± 2.82	6.27 ± 2.26	−0.34 (possibly deleterious)
			% Tactical Efficiency	81.35 ± 20.13	84.95 ± 17.45	0.31 (possibly beneficial)
			Errors	1.29 ± 1.49	0.93 ± 1.11	−0.41 (likely beneficial)
	Defensive Coverage	ICP	TPI	35.53 ± 21.45	25.98 ± 20.95	−0.99 (very likely deleterious)
			Frequency	2.13 ± 1.61	1.32 ± 1.10	−1.02 (very likely deleterious)
			% Tactical Efficiency	76.27 ± 33.79	63.68 ± 44.70	−0.64 (likely deleterious)
			Errors	0.39 ± 0.85	0.22 ± 0.47	−0.31 (possibly) beneficial
	Defensive Balance	OCP	TPI	29.87 ± 20.88	30.93 ± 21.02	0.11 (unclear)
			Frequency	2.61 ± 3.44	2.56 ± 3.21	−0.02 (unclear)
			% Tactical Efficiency	72.54 ± 40.97	77.24 ± 37.66	0.17 (unclear)
			Errors	0.31 ± 0.65	0.27 ± 0.64	−0.09 (unclear)
	Concentration	OCP	TPI	32.79 ± 14.58	31.84 ± 18.57	−0.17 (unclear)
			Frequency	4.99 ± 3.09	3.88 ± 2.80	−0.61 (likely deleterious)
			% Tactical Efficiency	87.14 ± 26.19	86.07 ± 29.56	−0.08 (unclear)
			Errors	0.41 ± 0.79	0.29 ± 0.71	−0.28 (possibly beneficial)
	Defensive Unity	OCP	TPI	31.08 ± 5.08	31.12 ± 5.17	0.01 (unclear)
			Frequency	27.22 ± 9.08	26.72 ± 8.30	−0.08 (unclear)
			% Tactical Efficiency	88.01 ± 10.52	90.39 ± 8.11	0.38 (likely beneficial)
			Errors	3.24 ± 2.93	2.45 ± 2.15	−0.43 (likely beneficial)

Note: % tactical efficiency: percentage of tactical efficiency; MBI: magnitude-based inference; errors: tactical errors; ICP: inside the center of play; ME ± SD: mean and standard deviation; OCP: outside the center of play; SSG: small-sided game; TPI: tactical performance index.

## Data Availability

The data presented in this study are available on request from the corresponding author.

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
