# Peer review of "The Effect of Acute Caffeine Ingestion on Tactical Performance of Professional Soccer Players"

_nutrients, 2022, doi:10.3390/nu14071466_

Round 1
Reviewer 1 Report
This study addresses caffein ingestion (CAF) on several aspects of tactical performance of professional soccer players (in small sided games). Overall, CAF lead to an increased ball possession time but failed to improve several individual offensive and defensive core principles. The article is very well organized and provides clear and adequate information on all sections. Text is concise but covers everything readers must understand about the subject and experiment. Limitations are addressed by the authors and they highlight how and why they are relevant.
In a more detailed analysis, I would like the make a few remarks. While phrasing seems to be on point, the are still a few minor mistakes: line 51 “as well low order cognitive functions”; line 56 “payers“; line 306 “However, this study present limitations“; line 319 “may are due”; line 327 “nutritionists may should”; 331 “should be interpret with”; line 333 “this aspect must also be considered for the use of caffeine by soccer players to be considered”;
Additionally, Figure 2 should follow the same color selection for “placebo” and “caffein” as in the other figures, for clearer and normalized visual presentation;
We don’t know exactly why 24 of the original 47 recruited subjects were requested by the club (or had other commitments). Were they “better” players than the rest of the sample? Were they older? More experienced? Does the final selection proportionally represent all field positions? Can this make any difference? Please add one or two paragraphs about this issue.
In summary, with this novel take on CAF, this study opens a new area of research, directly related to how CAF translates to soccer match play. We are now going beyond the well-known effects of CAF on physical performance. While results were somehow mixed, future studies should extend our comprehension of such aspects of CAF. Furthermore, and most importantly, they might change current paradigms of CAF on sports with a strong tactical component, by enlightening what kind of players (or field position) would clearly benefit with CAF, from the ones who wouldn’t.
As for me, I’m excited to see what kind of data future research will bring.
Author Response
We thank the reviewer for their time and constructive comments. We have made the amendments, but below is a summary of our responses...
This study addresses caffeine ingestion (CAF) on several aspects of tactical performance of professional soccer players (in small sided games). Overall, CAF lead to an increased ball possession time but failed to improve several individual offensive and defensive core principles. The article is very well organized and provides clear and adequate information on all sections. Text is concise but covers everything readers must understand about the subject and experiment. Limitations are addressed by the authors and they highlight how and why they are relevant.
The authors would like to thank the reviewer for taking the time to provide constructive feedback on our manuscript. We are pleased that you see the value of our work and in line with your suggestions have improved our submission to a standard which we now hope you will deem worthy of publication.
In a more detailed analysis, I would like the make a few remarks. While phrasing seems to be on point, the are still a few minor mistakes: line 51 “as well low order cognitive functions”; line 56 “payers“; line 306 “However, this study present limitations“; line 319 “may are due”; line 327 “nutritionists may should”; 331 “should be interpret with”; line 333 “this aspect must also be considered for the use of caffeine by soccer players to be considered”;
Thank you for the important observations, we have changed in the manuscript.
Additionally, Figure 2 should follow the same color selection for “placebo” and “caffein” as in the other figures, for clearer and normalized visual presentation;
Thank you for the observation and pertinent comment. We have adjusted figure 2 accordingly.
We don’t know exactly why 24 of the original 47 recruited subjects were requested by the club (or had other commitments). Were they “better” players than the rest of the sample? Were they older? More experienced? Does the final selection proportionally represent all field positions? Can this make any difference? Please add one or two paragraphs about this issue.
The final sample was composed of starting and reserve players. Not all starters were removed by club request, so considering that starters are the best players in each position, in the end we had a mixed sample. The reasons given by the club were diverse, for example players with a history of injury who, despite having recovered, were asked not to participate just as a precaution. And some others were asked to do other activities without the club providing us much information on the reason.
We added a small explanation in the manuscript (Page 2, lines 100-105) mentioning that the final sample was composed by starters and reserves to make this issue clearer, but we chose to keep it brief as the methods section is quite extensive. We hope that the changes made are in accordance with your suggestion, which is indeed very relevant.
In summary, with this novel take on CAF, this study opens a new area of research, directly related to how CAF translates to soccer match play. We are now going beyond the well-known effects of CAF on physical performance. While results were somehow mixed, future studies should extend our comprehension of such aspects of CAF. Furthermore, and most importantly, they might change current paradigms of CAF on sports with a strong tactical component, by enlightening what kind of players (or field position) would clearly benefit with CAF, from the ones who wouldn’t.
As for me, I’m excited to see what kind of data future research will bring.
Thank you for sharing our enthusiasm for this challenging but novel study
Reviewer 2 Report
Overall: The study aimed to investigate the effects of caffeine ingestion on tactical performance during small-sided games (SSG) in professional soccer players. The manuscript presented some concerns that preclude the approval of the presented text. Therefore, I have major and minor comments to improve the manuscript.
# Abstract:
- Page 1, Line 25: Remove the word “rare”.
- Page 1, Line 25: Remove the word “uniquely”.
- Page 1, Line 31: What does 3+GK means? Write in the unabbreviated form, please.
- Page 1, Line 32: Change “accessed” to assessed. Please review this typing error all over the manuscript.
- Page 1, Line 30: Is the design crossover? Because the explanation about the caffeine or placebo ingestion suggests that the design is crossover. Please adjust properly.
- Page 1, Line 34-35: Remove word “core” that was duplicated (“core offensive and defensive core”). Please adjust properly.
- Page 1, Line 37: Remove the word “overall” .
- Page 1, Line 42: Change the word “harmful” to deleterious.
# Introduction:
- Page 2, Line 50: Put a comma (,) after intermittent: “intermittent, and”. Please review the comma insertion all over the manuscript.
- Page 2, Line 63: Remove the comma (,) after “of ,”.
- Page 2, Line 66: Remove “as such”.
- Page 2, Line 67: Change “in game” to in-game.
- Page 2, Line 69: I believe there is a dot (.) after “et al”. Please adjust properly over the text.
- Page 2, Line 72: Are the Del Coso results compare with Control or placebo? Please insert this information with the p-values in the text.
- Page 2, Line 72-75: This sentence is too long; please rewrite it to make the information clear to the readers.
- Page 2, Line 76: I suggest to include the follow reference as well (Ferreira et al., 2021):
doi: 10.1177/1941738121998712.
- Page 2, Line 77: Remove the word “uniquely”.
- Page 2, Line 78: Remove the word “the”.
# Materials and Methods:
- Page 2, Line 84: Change “5,7” to 5.7
- Page 2, Line 85: Please provide a reference to this sentence “based on an estimated observed power (1-β) of 0.85” or correct it properly.
- Page 2, Line 86: Is the calculated sample size of 10 participants the total individuals needed for the study or for each group (total of 20 participants)? Please clarify this information in the text for the readers.
- Page 2, Line 87-90: As recommendation of CONSORT statement, please provide the Figure of the “flow diagram” of the progress through the phases of a parallel randomised trial of two groups (that is, enrolment, intervention allocation, follow-up, and data analysis). More information on CONSORT website.
- Page 2, Line 95: “Brazilian” National Health Council.
- Page 2, Line 97: Is the study parallel or crossover? Please include this information in the design description. In adittion, the informations regarding the experimental design are different among abstract and methods section (Abstract: “randomized, balanced, and double-blind clinical trial;” Methods: “randomized, within-subjects, repeated-measures, and double-blinded controlled design”). Please provide this information in a more simple and habitual description, eg.: randomized, double-blind, placebo-controlled and parallel-group clinical trial.
- Page 2, Line 98: How many familiarization sessions were performed? One, two; please make this information clear.
- Page 3, Line 101-102: Change “Participants then” to: Then, the participants
- Page 3, Line 104: Regarding the sentence “Solutions were identical in flavor and color”: Was any assessment performed to analyze whether the solutions were indeed identical in taste and color (as a placebo drink should be) with the participants?
- Page 3, Line 104: Change “accessed” to “assessed”. Review this typing error all over the manuscript, please.
- Page 3, Line 105: Which data was obtained from Maughan [14]? How were these data applied to assess caffeine consumption? Please clarify this information in the text.
- Page 3, Line 109: Change “but to” to “and”.
- Page 3, Line 142: In the sentence “maximum and during the games, no feedback was provided”, I suggest “maximum during the games, and no feedback was provided”.
- Page 3, Line 144: Put a comma (,) before "a defender".
- Page 4, Line 159 and 164: Please check reference used (Costa et al., 2011b; and 21).
# Results:
- Page 5, Line 191 (Figure 1 legend): “c) P < 0.05 vs. SSG3”, it should be “d)”.
- Line 242: Change “On the other hand” to a more elegant adverb: “conversely”.
# Discussion:
- Line 256: Remove the word “uniquely”.
- Line 259: Remove one word “core” cause is duplicated.
- Please discuss more why CAF presented better scores in some variables while PLA performed better in other parameters. This is pivotal to constructing a solid discussion section.
- Line 321-322: Please, change “essential effects of the various ergogenic measures” to the specific subject of the study: “understanding of the caffeine ergogenic effects”.
- As the blinding process may be understood as likely occurred (lines 117-119), please discuss about the “placebo effect of caffeine” and the “caffeine expectancies effects”. Initially, I suggest the following references:
. Placebo: Saunders et al., 2017 - doi 10.1111/sms.12793
. Expectancy: Shabir et al., 2019 - doi 10.3390/nu11102289
Shabir et al., 2018 - doi 10.3390/nu10101528
# Conclusion:
- Line 324: Put a comma (,) after “SSG”.
- Line 327-330: “Coaches… defenders.”, this sentence should be moved to discussion section, because it's not a conclusion, but a discussion argument.
- Line 333-334: Remove “to be considered.”
Author Response
We thank the reviewer for their time and constructive comments. We have made the amendments, but below is a summary of our responses...
Reviewer 2
We thank the reviewer for the detailed and insightful review of our manuscript. Your comment have helped to improve the quality of our submission and we hope that you now deem our work worthy of publication.
# Abstract:
- Page 1, Line 25: Remove the word “rare”.
- Page 1, Line 25: Remove the word “uniquely”.
- Page 1, Line 31: What does 3+GK means? Write in the unabbreviated form, please.
- Page 1, Line 32: Change “accessed” to assessed. Please review this typing error all over the manuscript.
- Page 1, Line 30: Is the design crossover? Because the explanation about the caffeine or placebo ingestion suggests that the design is crossover. Please adjust properly.
- Page 1, Line 34-35: Remove word “core” that was duplicated (“core offensive and defensive core”). Please adjust properly.
- Page 1, Line 37: Remove the word “overall” .
- Page 1, Line 42: Change the word “harmful” to deleterious.
Thank you for the comments above. We updated the manuscript accordingly.
# Introduction:
- Page 2, Line 50: Put a comma (,) after intermittent: “intermittent, and”. Please review the comma insertion all over the manuscript.
- Page 2, Line 63: Remove the comma (,) after “of ,”.
- Page 2, Line 66: Remove “as such”.
- Page 2, Line 67: Change “in game” to in-game.
- Page 2, Line 69: I believe there is a dot (.) after “et al”. Please adjust properly over the text.
- Page 2, Line 72: Are the Del Coso results compare with Control or placebo? Please insert this information with the p-values in the text.
We updated the manuscript accordingly. As the authors did not present the exact p value in their paper, we added information as "(P < 0.05)".
- Page 2, Line 72-75: This sentence is too long; please rewrite it to make the information clear to the readers.
- Page 2, Line 76: I suggest to include the follow reference as well (Ferreira et al., 2021):
doi: 10.1177/1941738121998712.
- Page 2, Line 77: Remove the word “uniquely”.
- Page 2, Line 78: Remove the word “the”.
Thank you for the comments above. We updated the manuscript accordingly.
# Materials and Methods:
- Page 2, Line 84: Change “5,7” to 5.7
- Page 2, Line 85: Please provide a reference to this sentence “based on an estimated observed power (1-β) of 0.85” or correct it properly.
To the best of our knowledge, there is no consensus on the best power to use in the determination of sample size in studies with sport science, although there are those who claim that "The ideal power for any study is considered to be 80% (https: //www.ncbi.nlm.nih.gov/pmc/articles/PMC3409926/). Similarly, Jones et al (https://emj.bmj.com/content/20/5/453) mentioned that "There is less convention as to the accepted level of pβ, but figures of 0.8–0.9 are common". For this reason, we used a (1-β) of 0.85 for the determination of minimum sample size. We added a sentence in the manuscript to justify our approach (Page 2, lines 99-100).
- Page 2, Line 86: Is the calculated sample size of 10 participants the total individuals needed for the study or for each group (total of 20 participants)? Please clarify this information in the text for the readers.
Thank you for your comment. The sample calculation resulted in a minimum of 10 participants per condition. According to the availability of athletes, we used 19 participants who were tested in both conditions. We have now clarified this in the section 2.1. Subjects, and we hope that the inclusion of the flow diagram (new Figure 1) as suggested further clarifies this.
- Page 2, Line 87-90: As recommendation of CONSORT statement, please provide the Figure of the “flow diagram” of the progress through the phases of a parallel randomised trial of two groups (that is, enrolment, intervention allocation, follow-up, and data analysis). More information on CONSORT website.
We appreciated the suggestion. A flow diagram based on CONSORT recommendations has been added (new Figure 1).
- Page 2, Line 95: “Brazilian” National Health Council.
- Page 2, Line 97: Is the study parallel or crossover? Please include this information in the design description. In adittion, the informations regarding the experimental design are different among abstract and methods section (Abstract: “randomized, balanced, and double-blind clinical trial;” Methods: “randomized, within-subjects, repeated-measures, and double-blinded controlled design”). Please provide this information in a more simple and habitual description, eg.: randomized, double-blind, placebo-controlled and parallel-group clinical trial.
- Page 2, Line 98: How many familiarization sessions were performed? One, two; please make this information clear.
- Page 3, Line 101-102: Change “Participants then” to: Then, the participants
- Page 3, Line 104: Regarding the sentence “Solutions were identical in flavor and color”: Was any assessment performed to analyze whether the solutions were indeed identical in taste and color (as a placebo drink should be) with the participants?
- Page 3, Line 104: Change “accessed” to “assessed”. Review this typing error all over the manuscript, please.
- Page 3, Line 105: Which data was obtained from Maughan [14]? How were these data applied to assess caffeine consumption? Please clarify this information in the text.
- Page 3, Line 109: Change “but to” to “and”.
- Page 3, Line 142: In the sentence “maximum and during the games, no feedback was provided”, I suggest “maximum during the games, and no feedback was provided”.
- Page 3, Line 144: Put a comma (,) before "a defender".
- Page 4, Line 159 and 164: Please check reference used (Costa et al., 2011b; and 21).
Thank you for the comments above. We updated the manuscript accordingly.
# Results:
- Page 5, Line 191 (Figure 1 legend): “c) P < 0.05 vs. SSG3”, it should be “d)”.
- Line 242: Change “On the other hand” to a more elegant adverb: “conversely”.
# Discussion:
- Line 256: Remove the word “uniquely”.
- Line 259: Remove one word “core” cause is duplicated.
Thank you for the remarks. We updated the manuscript accordingly.
- Please discuss more why CAF presented better scores in some variables while PLA performed better in other parameters. This is pivotal to constructing a solid discussion section.
We agree with the observation. Indeed, a discussion of why CAF resulted in improvements in some variables while not in others is pertinent. However, this is a great challenge, since the technical-tactical variables analysed herein involve several different cognitive and technical factors, and it is often difficult to discriminate the underpinnings of each tactical principle. We are aware that this is a limitation of the study, as pointed out in the discussion. A strength in the methods used is the great ecological validity, but the technical-tactical analysis tool used makes it difficult to relate to specific mechanisms of caffeine without being excessively speculative.
We have tried to make it clear in the discussion that these limitations exist, but we hope that this study will prompt further investigation into the effects of caffeine (and other supplements) on decision-making in football players. We have also added more information in the discussion (as highlighted with the track changes function).
- Line 321-322: Please, change “essential effects of the various ergogenic measures” to the specific subject of the study: “understanding of the caffeine ergogenic effects”.
Thank you for the remarks. We updated the manuscript accordingly.
- As the blinding process may be understood as likely occurred (lines 117-119), please discuss about the “placebo effect of caffeine” and the “caffeine expectancies effects”. Initially, I suggest the following references:
. Placebo: Saunders et al., 2017 - doi 10.1111/sms.12793
. Expectancy: Shabir et al., 2019 - doi 10.3390/nu11102289
Shabir et al., 2018 - doi 10.3390/nu10101528
Thank you for your suggestion. It is indeed a very pertinent topic, so we added more information about this int he last paragraph of the discussion, when presenting the limitations of our study. Wes used two references from the suggested above.
# Conclusion:
- Line 324: Put a comma (,) after “SSG”.
- Line 327-330: “Coaches… defenders.”, this sentence should be moved to discussion section, because it's not a conclusion, but a discussion argument.
- Line 333-334: Remove “to be considered.”
Thank you for the remarks. We updated the manuscript accordingly.